# Role of Neuronal TRPC6 Channels in Synapse Development, Memory Formation and Animal Behavior

**DOI:** 10.3390/ijms242015415

**Published:** 2023-10-21

**Authors:** Nikita Zernov, Elena Popugaeva

**Affiliations:** Laboratory of Molecular Neurodegeneration, Peter the Great St. Petersburg Polytechnic University, 195251 St. Petersburg, Russia

**Keywords:** TRPC6, neuroprotection, memory, behavior

## Abstract

The transient receptor potential cation channel, subfamily C, member 6 (TRPC6), has been believed to adjust the formation of an excitatory synapse. The positive regulation of TRPC6 engenders synapse enlargement and improved learning and memory in animal models. TRPC6 is involved in different synaptoprotective signaling pathways, including antagonism of N-methyl-D-aspartate receptor (NMDAR), activation of brain-derived neurotrophic factor (BDNF) and postsynaptic store-operated calcium entry. Positive regulation of TRPC6 channels has been repeatedly shown to be good for memory formation and storage. TRPC6 is mainly expressed in the hippocampus, particularly in the dentate granule cells, cornu Ammonis 3 (CA3) pyramidal cells and gamma-aminobutyric acid (GABA)ergic interneurons. It has been observed that TRPC6 agonists have a great influence on animal behavior including memory formation and storage The purpose of this review is to collect the available information on the role of TRPC6 in memory formation in various parts of the brain to understand how TRPC6-specific pharmaceutical agents will affect memory in distinct parts of the central nervous system (CNS).

## 1. Introduction

Research of learning and memory mechanisms suggests that a continuous increase in the strength of synaptic transmission is necessary to achieve long-term modification of neural network properties and memory storage. In the hippocampus, most excitatory postsynaptic terminals are found in dendritic spines. The spine’s volume is closely related to the function: larger spines have a wider postsynaptic density, more functional α-amino-3-hydroxy-5-methyl-4-isoxazole propionic acid receptors (AMPARs), and likely elicit a larger excitatory postsynaptic potential.

Previously, it was suggested that mushroom spines formed between excited neurons are stable “memory spines” that form functionally strong synapses and are therefore responsible for memory storage [1]. Many groups of researchers suggest that the loss of mushroom spines may underlie the decline in cognitive functions of the brain during the progression of Alzheimer’s disease (AD) [2,3,4] and other neurodegenerative diseases (NDD).

The transient receptor potential cation channel, subfamily C, member 6 (TRPC6), appears to be essential for the formation of an excitatory synapse. The positive regulation of TRPC6 engenders synapse enlargement and improved learning and memory in animal models [5]. Even though the exact physiological role of TRPC6 is still contentious, the channel is critical to cognitive functions. A vast variety of central nervous system (CNS)-related diseases such as epilepsy, autism spectrum disorder and AD are characterized by cognitive impairments and may be provoked by TRPC6 malfunctions [6,7,8,9]. 

The calcium hypothesis of AD states that the disruption of Ca^2+^ homeostasis through the abnormal functioning of calcium-permeable proteins, such as plasma membrane ion channels (N-methyl-D-aspartate receptor (NMDAR), AMPAR, voltage-gated calcium channels (VGCC)), ion channels of endoplasmic reticulum (Ryanodine receptor (RyanR), inositol trisphosphate receptor (IP3R), store-operated channels, the mitochondrial permeability transition pore (mPTP), underlies the pathogenesis of AD [10,11,12]. In confirmation of the functionality of the calcium hypothesis, the only pharmacological drug that temporarily alleviates the symptoms of AD is the NMDAR blocker—memantine [13]. According to the calcium hypothesis, calcium channels, such as TRPC6, may be a promising avenue for the development of pharmacological solution of AD.

Cognitive dysfunctions and synaptic degeneration are related in AD patients’ brains. Agents that reduce synapse loss are possible pharmacological solutions for AD treatment. TRPC6-positive modulators have repeatedly been shown to slow down spine loss, thus, they may be considered as candidates for NDD’s treatment [14]. There is some genetic confirmation of TRPC6’s role in AD pathogenesis. TRPC6 mRNA levels in the blood cells [15,16] are specifically reduced in patients with AD and moderate cognitive impairment as well as in AD patient-specific induced pluripotent stem cells (iPSCs) [17]. The TRPC6 overexpression or the pharmacological positive stimulation of them recovers store-operated calcium entry (nSOCE) in hippocampal neurons in AD [18,19]. Overexpression of TRPC6 in the mouse brain is correlated with cognitive recovery and excitatory synapse enlargement [5]. TRPC6 channels have been identified as key molecular players of nSOCE in the hippocampus [18]. It has been found that the knockdown of TRPC6 expression obstructs nSOCE [18]. It has been shown that the long-term potentiation deficit in brain slices taken from AD transgenic mouse models can be recovered by TRPC6 agonists [19,20].

Derivative of benzopyran (C20), has been recently revealed as a novel selective TRPC6 positive modulator [21]. We have previously investigated its therapeutic profile and suggest that C20 might be recognized as a perspective prototype of a pharmacological agent that is able to reduce cognitive decline [20]. The purpose of this review is to collect the available information on the role of TRPC6 in memory formation in various parts of the brain, in order to understand how the TRPC6-specific pharmaceutical agent will affect memory in other parts of the CNS.

## 2. TRPC6 as Calcium Permeable Channel

The transient receptor potential (TRP) channels superfamily includes a diverse group of cation channels that are highly conserved from Drosophila to mammals. Based on sequence homology the mammalian TRP channel superfamily is divided into several subfamilies, the first of which constitutes the “canonical” TRP subfamily (TRPC). TRPC channels can be further classified into four different subfamilies: TRPC1 (I), TRPC2 (II), TRPC3, 6, 7 (III) and TRPC4, 5 (IV). Members of the III subfamily (TRPC3, 6 and 7) share a high degree of amino acid identity (approximately 70–80%) [22,23,24]. TRPC3 and TRPC7 have slightly greater amino acid identities to each other than TRPC6 [22,23,24]. Another difference between these channel types is their ion permeability. TRPC6 is observed to be more calcium-selective, while TRPC3 and TRPC7 do not appear to be. The reported ion permeability ratio PCa/PNa for TRPC6 is 5 [25], while the values for TRPC3 and TRPC7 are 1.5 and 2, respectively [26].

The putative transmembrane structure is similar to that of other TRP channels. TRPC6 features six transmembrane-spanning helices with intracellular N- and C-termini and a pore-forming loop. TRPC6 monomers form a TRPC tetramer with a functional pore domain in the center [27]. 

The mechanism of TRPC6 activation is debatable but seems that both receptor-operated channel (ROC) activation and store-operated channel (SOC) activation interplay in a cell machinery. ROC: G protein-coupled receptor-activated phospholipase C (PLC) can modulate TRPC6 channel activity by hydrolysis of phosphatidylinositol bisphosphate (PIP2) to diacylglycerol (DAG) and inositol trisphosphate (IP3) [25]. DAG can directly activate TRPC6. Interestingly, the binding of brain-derived neurotrophic factor (BDNF) to tropomysin-related kinase B (TrkB) results in the activation of PLC which also leads to the TRPC6 activation [28]. SOC: IP3 may bind to IP3 receptors (IP3R) which leads to the calcium release from the main intracellular Ca^2+^ store (the endoplasmic reticulum). IP3-mediated emptying of the endoplasmic reticulum triggers TRPC6 activation via a store-dependent mechanism [29,30] (Figure 1).

## 3. TRPC6-Mediated Synapse Development

TPRC6 has been identified as an essential component for excitatory synapse formation. Overexpressing TRPC6 greatly increased dendritic spine density and the level of synapsin1 and PSD-95 cluster, known as the pre- and postsynaptic markers [5]. Moreover, overexpressing TRPC6 enhanced phosphorylation of Cyclic adenosine monophosphate (cAMP) response element binding protein (CREB) and Calcium/calmodulin-dependent protein kinase IV (CaMKIV) [5]. Expressing a dominant-negative form of either CREB or CaMKIV repealed TRPC6-induced spine formation which suggests that CAMKIV/CREB pathway seems to be crucial for the formation of spines and excitatory synapses [5,31]. Please, see more about role of CREB in TRPC6-mediated signaling pathways in Section 4.

CAMKII are also important for TRPC6-mediated synapse development. It is suggested that store-operated calcium entry (SOCE) via TRPC6 is needed to activate (CaMKII). CaMKII that undergoes autophosphorylation and in turn phosphorylates GluR1 (subunits of AMPAR) [32]. This promotes AMPARs trafficking to the post-synaptic membrane [33,34], which increases membrane depolarization and enhances NMDAR’s contribution to LTP maintenance (Figure 1). Simultaneously, structural changes in the synapse, which strengthen the synaptic connection, occur. Since the calcium permeability for TRPC6 is significantly lower than for NMDAR and TRPC6-SOCE is not voltage-dependent, the role of TRPC6-SOCE in maintaining spines plays a particularly important role in silent synapses [3,18]. It has been also shown that the knockdown of CaMKII in hippocampal dendritic spines impairs neuronal SOCE [35]. The observed effect is likely due to CaMKII’s involvement in TRPC6 phosphorylation [36,37] and subsequent regulation of its Ca^2+^ permeability.

## 4. TRPC6-Mediated Neuroprotective Intracellular Signaling Pathways

Accumulating evidence from a variety of experimental models suggests that TRPC6 participates in neuroprotection. TRPC6 overexpression has been shown to rescue mushroom spine loss in presenilin and amyloid precursor protein (APP) knock-in mouse models of AD [18] and also protect neurons from ischemic brain damage [31]. Mice that overexpress TRPC6 in the brain show improved cognitive function and increased excitatory synapse formation [5]. TRPC6-dependent protective signaling pathways differ between distinct NDD. Below we describe TRPC6 involving signaling pathways that take place in certain NDD and try to explain whether it takes place in AD. In Section 4, we described TRPC6-mediated intracellular signaling pathways that may lead to neuroprotection at the cellular level regardless of whether this path is intracellular or not.

TRPC6’s neuroprotective effect could be explained due to the antagonism of extrasynaptic NMDAR-mediated intracellular calcium overload. Hyperactivation of NMDAR is a critical event in glutamate-driven excitotoxicity that causes a rapid increase in intracellular calcium concentration [38]. Such rapid increases in cytoplasmic calcium concentrations may activate and over-stimulate a variety of proteases, kinases, endonucleases, etc. This downstream neurotoxic cascade may trigger severe damage to neuronal functioning [39]. Hyperactivation of NMDAR is frequently observed during brain ischemia when a huge number of neurons die due to acute oxygen deprivation [31]. In contrast to acute brain ischemia, AD is a chronic disease. Neurons are dying in AD brain but not as fast as in brain ischemia, thus such TRPC6-dependent antagonism of NMDAR hyperactivation might take place at late stages of AD when a significant amount of neurons are lost. Indeed, memantine, the inhibitor of extrasynaptic NMDAR [40], demonstrates a similar neuroprotective effect in middle to advanced forms of AD. NMDA receptors are composed of two obligatory NR1 subunits and two regulatory NR2A or NR2B subunits. The ratio of NR2A/B NMDA receptor subunits may change during postnatal development. In the adult cortex, extrasynaptic NMDAR preferentially contains NR2B subunit over NR2A [41,42,43]. Interestingly, inhibition of NR2A-containing NMDA receptors increases neuronal death after transient global ischemia, but inhibition of NR2B-containing NMDA receptors reduces ischemic neuronal death [44] which may reflect different roles of NMDAR-dependent intracellular signaling pathways in synaptic and extrasynaptic locations. Activation of NR2B-containing NMDARs enhances TRPC6 degradation through calpain [45]. Furthermore, TRPC6 overexpression has been reported to inhibit the NMDAR-dependent calcium elevation as assayed by calcium imaging [31]. Moreover, the whole-cell patch-clamp recording shows that overexpression or pharmacological activation of TRPC6 selectively suppresses the NMDAR-dependent current [46]. Conversely, the downregulation of TRPC6 enhances this current [46] and aggravates calcium elevation [31]. The possible mechanism by which TRPC6 may impede the NMDAR activity is their dephosphorylation by calcineurin. Calcineurin may be activated by TRPC6. It has been demonstrated that a calcineurin inhibitor reversed the TRPC6 inhibition of NMDAR-dependent current [31]. This suggested that TRPC6 may impede the NMDAR activity via calcineurin. Thus, the neuroprotective effect may be achieved by suppressing the toxic calcium entry through extrasynaptic NR2B-containing NMDAR (see dashed lines in Figure 1).

TRPC6-dependent neuroprotection has been also connected with agonism of synaptic NMDAR. For example, activation of NR2A-containing NMDAR increased TRPC6 mRNA synthesis through a calcineurin/nuclear factor of activated T-cells (NFAT)-dependent pathway in cortical neurons (see solid lines in Figure 1) [45]. Agonism of synaptic NMDAR has been proposed to underlie the activation of synaptoprotective store-operated calcium entry via TRPC6 in hippocampal silent synapses. TPRC6 has been identified as a key component of store-operated calcium entry (SOCE) in hippocampal neurons [18]. SOCE is a ubiquitous signaling module connecting the ER, with plasmalemmal calcium entry [11,47]. Upon depletion of ER, the stromal interaction molecule (STIM) detects calcium level reduction and activates Orai channel proteins and/or TRPC underlying SOCE [18,47]. It is suggested that Ca^2+^ entry via TRPC6-neuronal SOCE is needed to activate CaMKII. CaMKII phosphorylates AMPAR [32,48], which promotes AMPARs trafficking to the post-synaptic membrane [33,34]. This increases NMDAR’s contribution to LTP maintenance (Figure 1). Furthermore, TRPC6-dependent neuronal SOCE is required for neuroprotection from amyloid’s and mutant presenilins’ toxic effects in vitro. The knockdown of TRPC6 expression hinders SOCE in primary hippocampal culture. The overexpression of TRPC6 channels or their positive stimulation restores SOCE and the loss of spines in hippocampal neurons in AD models [18,19].

Cyclic adenosine monophosphate (cAMP) response element binding protein (CREB) plays a pivotal role in neuronal survival. The phosphorylation of CREB has been involved in a large range of neuroprotective signaling pathways [49]. In PC12 cells and primary hippocampal neurons, calcium influx through TRPC6 may activate CREB through three signal cascades: Ras/MEK/ERK, RAS/PI3K/Akt, and CaM/CAMKIV [5,50]. It has been demonstrated that a prominent TRPC6 modulator, hyperforin, causes the activation of a GTPase called Ras (from “Rat sarcoma virus”). Activated Ras activates a RAF kinase (RAF is an acronym for “Rapidly Accelerated Fibrosarcoma”). The RAF kinase may activate the lipid kinase phosphatidylinositol-3-kinase (PI3K). PI3K phosphorylates and activates AKT (also known as protein kinase B), which leads to the phosphorylation of CREB. Another effector for Ras is MAPK/ERK Kinase (MEK). The MEK phosphorylates a mitogen-activated protein kinase (MAPK), originally called extracellular signal-regulated kinase (ERK). Activation of MAPK likewise results in CREB phosphorylation [51]. The inhibition of PI3K/Akt or ERK/MEK pathways has been determined to constrain hyperforin-mediated CREB phosphorylation [52]. CAMKII and IV are also possible downstream targets for Ras. Pharmacological activation or overexpression of TRPC6 promotes CAMKIV and CREB phosphorylation [5,50,52] (Figure 1). CAMKIV/CREB pathway seems to be crucial for the formation of spines and excitatory synapses [5,50]. 

It is known that the CREB is activated in neurons in response to a diverse array of stimuli, including growth factors such as brain-derived neurotrophic factor (BDNF) BDNF is a neurotrophin, essential for the central nervous system’s development, survival, and neuronal plasticity. The reduction of the BDNF level is observed in several neurodegenerative disorders, including Huntington’s disease, AD, and Parkinson’s disease [53]. TRPC6 is also required for the neuroprotective effect of BDNF. Downregulation of TRPC6 attenuates the neuroprotective effect of BDNF-induced intracellular calcium elevation, and CREB activation on cerebellar granule cells [28]. The promoter region of BDNF contains CRE; activated CREB binds to CRE and promotes the transcription of BDNF [54] (Figure 1). Interestingly, the other study demonstrated that pharmacological blocking of TRPC6 channels in rat retina leads to a significant increase in proBDNF (the BDNF precursor), while the level of mature BDNF (mBDNF) remains nearly constant [55]. Elevation of proBDNF may be suggested as feedback to the BDNF-initiated protective mechanism in the response of TRPC6 blocking.

Another potential downstream target for TRPC6-induced calcium entry is the biosynthesis of the two most common endocannabinoids, N-arachidonylethanolamine (anandamide, AEA) and 2-arachidonyl glycerol (2-AG) (Figure 1). 2-AG and AEA are synthesized in a calcium-dependent manner. Both ligands are produced “on-demand” in response to increased calcium concentrations [56]. OAG-induced TRPC6 activation has been discovered to promote 2-AG and AEA biosynthesis in the catecholaminergic neuronal tumor CAD cells (Cath.-a-differentiated). Besides, the knockdown of TRPC6 repressed OAG-stimulated endocannabinoid synthesis [57]. A growing body of evidence suggests that the anti-inflammatory properties of endocannabinoids are important for neuroprotection [58,59,60,61]. Particularly, the anti-inflammatory effects of 2-AG are possibly mediated through cannabinoid receptors 1 and 2 (CB1/2)-dependent and -independent mechanisms. Activation of the peroxisome proliferator-activated receptor-γ (PPARγ) appears to be a critical component in 2-AG-modulated CB1/2-independent neuroprotection. Presumably, the neuroprotective effects are achieved by suppressing nuclear factor-κB (NF-κB) [59,62].

TRPC6-dependent regulation of BDNF and endocannabinoids, has not been precisely studied in AD models, thus it is possible that these intracellular signaling pathways provide neuroprotective effects of described earlier TRPC6 agonists (including C20) in AD affected brains.

## 5. TRPC6 Dependent Signaling Pathways in Hippocampal DGC and GABAergic Interneuron

Although TRPC6 expression in the brain is lower than that of other TRPCs, its mRNA is detected in many different areas of the central neuronal system [63]. TRPC6 have been shown to be expressed in cerebellum, cortex, middle frontal gyrus, hippocampal pyramidal neurons, interneurons, cortical astrocytes [64,65,66]. TRPC6 is predominantly expressed in the hippocampus and cerebrum [18]. Particularly, TRPC6 is specifically expressed in hippocampal dentate granule cells (DGC), CA3 pyramidal cells and GABAergic interneurons [67]. In Section 5, we attempt to emphasize the role of TRPC6 in hippocampal DGC and GABAergic interneurons in order to demonstrate possible supracellular mechanism of neuronal regulation.

A recent study has shown a significant effect of shRNA-mediated knockdown of TRPC6 in hippocampal dentate gyrus (DG) on animal behavior [68]. Particularly, the shRNA-TRPC6 treated mice demonstrated spatial learning and memory abnormalities. Authors have identified in DGCs loss of synapses, decrease in PSD95, changes in pAkt and CREB expression as underlying signaling pathways that might cause observed cognitive dysfunctions [68]. In agreement with the literature, morphological synaptic abnormalities that occurred in DGCs and cognitive deficiency are correlated in human and mice. The DG is responsible for a variety of cognitive tasks, for example pattern separation, pattern completion, novelty detection, etc. [69,70,71,72]. Interestingly, knockdown of TRPC6 altered mouse nest building behavior and spontaneous alternation behavior on Y maze, which means that downregulation of TRPC6 obstructs short spatial memory. Contrariwise, memory was not hindered in the new object recognition test and in the Morris water maze. However, an escape latency time was extended in condition of TRPC6 downregulation which is assumed mild spatial memory impairments. In the open field test, the shRNA TRPC6 treated more active and tend to avoid the central area [68]. Taking together these behavioral data indicates an important role of TRPC6 in hippocampal memory acquisition and its transformation to the specific behavior.

The entorhinal cortex provides excitatory neuron input to DGC, which then give excitatory output to the hippocampus CA3 area through mossy fibers. Additionally, this area contains a variety of GABAergic interneurons that modulate granule cell activity via feedback and feedforward inhibition [73]. TRPC6 has been shown to regulate GABAergic interneuron inhibitions onto the DGC and CA1 pyramidal cells during and after high frequency stimulation (HFS). TRPC6 knockdown reduces the voltage-gated potassium channel 4.3 (Kv4.3) translocation and its dendritic localization of DGC and GABAergic interneurons. Kv4.3 restricts the back-propagation of action potentials into the dendrites and decreases excitatory synaptic events [74]. Downregulation of these channels impedes fast-spiking in interneurons and afterwards increases neuronal excitability in principal neurons in multiple ways [75,76]. Furthermore, TRPC6 knockdown also inhibits ERK1/2 activity in DGC and GABAergic interneurons. Remarkably, TRPC6 knockdown’s effects were reversed by a hyperforin [77]. These results offer an evidence that TRPC6 may have a significant impact on the maintenance of neuronal excitability via ERK1/2-mediated membrane Kv4.3 localizations in the DG. TRPC6-ERK1/2 neuroprotective signaling pathway (protection from prolonged seizure activity) has been also reported to protect DGC via facilitation of mitochondrial fission [78].

## 6. Influence of TRPC6 Agonists on Animal Behavior Including Memory Formation and Storage

The impact of TRPC6 on behavior cannot be denied. TRPC6 transgenic mice show improved hippocampus-dependent spatial memory in the Morris water maze paradigm [5] and better motor performance after ischemia in the rotarod test [31]. Whereas, TRPC6 knockdown reduces exploration in the square open field and the elevated star maze but does not demonstrate any measurable variations in anxiety in the marble-burying test [79].

TRPC6 can be activated by a wide variety of agents, for instance, DAGs [25], lysophosphatidylcholines [80], 20-hydroxyeicosatetraenoic acid (an arachidonic acid’s metabolite) [81], different DAG analogs [82,83], and docosanoid neuroprotectin D1 [84]. Some synthetic agents (flufenamic acid [85], several pyrazolopyrimidines [45], piperazine derivatives [86], the benzimidazole-based GSK1702934A, its azobenzene derivative OptoBI-1 [87], and the benzopyran derivative C20 [21]) are structurally different from DAG but have also been identified as TRPC6 agonists. They are able to activate TRPC6 directly in receptor-operated mode. Interestingly, that the DAG serves as an obligate co-factor for the piperazine derivative, 51164, which means that 51164 is possible to stimulate TRPC6 in store-operated mode [19]. Contrary to synthetic agents, natural compounds (stilbenoid resveratrol [88], the isoflavone calycosin [89], and (−)-epigallocatechin-3-gallate, a catechin-type polyphenol [90], and the aminoquinazoline derivative, NSN21778) activate TRPC6 vicariously (store-operated mode). In this case, TRPC6-induced calcium entry is a response to intracellular calcium store depletion. Despite existence of wide range of TRPC6 targeting chemicals, only limited number of them have been tested in behavioral studies.

Hyperforin, the major antidepressant constituent of St. John’s wort, has been demonstrated as an effective pharmacological agent that recovers cognitive deficits in different behavioral tests. Treatment with hyperforin decreased the number of aggressive characteristics such as latency to the first attack, number of fights in isolation-induced aggression, and duration of water consumption, frequency of water spout possession in the water competition test [91]. Hyperforin likewise is able to overcome the memory impairments in the Morris water maze test [92], the open field test, novelty suppressed feeding test, the forced swimming test [93], conditioned avoidance, and passive avoidance tests [94]. However, hyperforin is unstable, difficult to synthesize [95], and demonstrates side effects and protonophore properties [96]. To overcome the disadvantageous chemical properties of hyperforin, different hyperforin analogs are developing rapidly. For example, Hyp13 may exhibit antidepressant and anxiolytic properties in the open field test, the novelty-suppressed feeding test, and the forced swimming test [97]. Another Hyperforin derivative IDN5706 recovers spatial cognitive damages in the Morris water maze test [98].

Some other TRPC6 activators have been demonstrated to improve cognitive impairments. For instance, the piperazine derivative AZP2006 enhances cognitive performance in the Y-maze test and the passive avoidance test [99]. Resveratrol, an indirect positive modulator of TRPC6 activity [88], attenuates behavioral impairment in rats with type 2 diabetes mellitus (T2DM) induced by a high-fat diet and streptozotocin in the novel object recognition test, elevated plus maze test, light–dark passive avoidance tests, radial arm maze and nest building behavior study [100].

A positive allosteric modulator of TRPC6, benzopyran derivative C20, efficiently recovers cognitive deficit in 6-month-old 5xFAD mice in the contextual and tone fear conditioning test [20]. Interestingly, it is believed that these two types of conditioning are related to different brain structures. Recognition of the context is dependent on the hippocampus, while recognition of an explicit stimulus (tone) is unaffected even in animals with hippocampal lesions [101] and mostly dependent on the function of the amygdala [102]. According to the human brain atlas (https://www.proteinatlas.org/ENSG00000137672-TRPC6/brain, accessed on 1 October 2022), TRPC6 is expressed in the amygdala. It is unclear if TRPC6 activation in the amygdala may impact tone-conditioned memory. However, it has been stated that TRPC5 channels are important for hippocampus-independent intrinsic fear memory [103]. In embryonic brain samples, TRPC5 may exist in a heterocomplex with TRPC6 [104]. It is unknown if the amygdala contains these heterocomplexes. Therefore, we can only speculate that observed C20-mediated pharmacological impact on TRPC6 is due to the potential modulation of TRPC6/TRPC5 complex in the hippocampus.

The impact of positive pharmacological modulation of TRPC6 on animal behavior and memory is summarized in Table 1.

## 7. Conclusions

At the cellular level, the functional unit of memory formation is believed to be a synaptic connection of neurons. The molecular machinery responsible for memory formation in synapses is considered to employ an intricate interplay of diverse biochemical processes that enables synaptic plasticity. Long-term synaptic plasticity is an activity-dependent change in neuronal connection strength over lengthy spans of time. It may manifest as LTP [112] or long-term depression [113,114,115]. However, it is obvious that memory is more than the LTP-LTD paradigm. Normal neuronal cells’ functioning is vital for all cognitive processes including memory formation. The fact of the positive effect of TRPC6 activators on memory is indisputable.

In the current review, we have described different molecular pathways that we have tried to combine in the unified scheme (Figure 1). However, we still lack a good understanding of how all these different components work together in order to provide a neuroprotection. TRPC6-dependent regulation of BDNF or endocannabinoids has not been precisely studied in AD models or at least in hippocampal neurons, thus their contribution to the positive influence of TRPC6 agonists in AD-affected brains remains controversial. In contrast to this, NMDA antagonism, CaMKIV-CREB or SOCE-CaMKII pathways have stronger arguments in favor of their participation in TRPC6-mediated neuroprotection.

However, the overactivation of TRPC6 may result in detrimental side effects. Excessive TRPC6 channel activation was demonstrated to cause glomerular damage due to their function in podocytopenia which all contribute to diabetic kidney disease (DKD) [116,117]. The TRPC6 channel’s gain-of-function mutations have been discovered as a possible hereditary cause of renal disorders such as focal segmental glomerulosclerosis [118]. Moreover, 2.4% of patients, who achieved therapy with 300 mg of hypericum extract 3 times a day, complained on some negative side-effects. Gastrointestinal irritations and allergy were the most frequently reported [119]. Thus, there is a need to pick up the lowest effective dose of the TRPC6-specific drug and to optimize the drug regime in order to reduce the side effects of TRPC6 positive modulation.

Neuroprotective TRPC6-dependent intracellular signaling mechanisms are present in different neuronal populations including glutamatergic exciting neurons and GABAergic inhibitory interneurons. Analyzing data summarized in Table 1 therapeutic effect of TRPC6 positive regulation is observed in distinct brain regions. Most of the literature repeatedly shows that upregulation of TRPC6 function has a positive effect on the function of the hippocampus and amygdala in different NDD-related diseases including AD. This is in agreement in terms of AD treatment, since the hippocampus is the first brain region to be affected by the disease. Thus, the usage of TRPC6 positive modulators would be most effective at early time points of the disease.

## Figures and Tables

**Figure 1 ijms-24-15415-f001:**
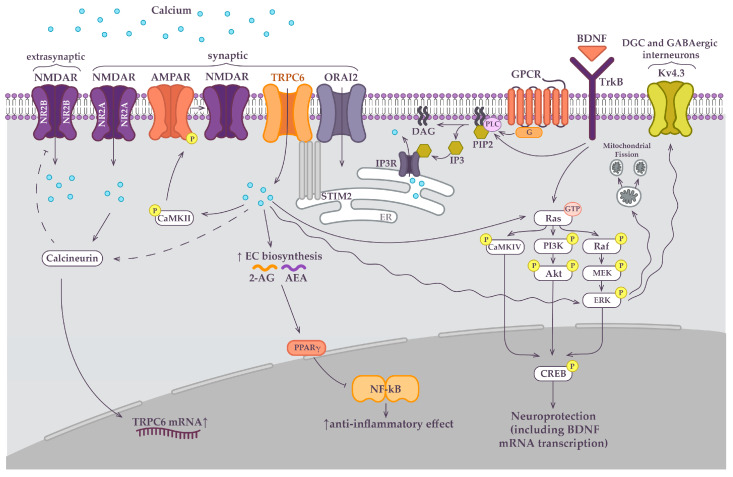
Neuroprotective intracellular signaling pathways in the central nervous system involving TRPC6 activation. A detailed description of pathways is given in the 3rd, 4th and 5th sections. Presumably, TRPC6 may impede the extrasynaptic NR2B-containing NMDAR-dependent calcium influx via calcineurin-involving dephosphorylation mechanism (see the dashed lines). On the other hand, activation of synaptic NR2A-containing NMDAR may increase TRPC6 mRNA synthesis through a calcineurin/nuclear factor of activated T-cells-dependent pathway in cortical neurons (see the solid lines). Agonism of synaptic NMDAR has been proposed to underlie the activation of synaptoprotective store-operated calcium entry via TRPC6 in hippocampal silent synapses. TPRC6 is key component of store-operated calcium entry in hippocampal neurons. TRPC6 may be activated directly either by DAG or pharmacological agent such as hyperforin as well as by store-depletion (following IP3R activation). Upon store depletion of ER, the STIM2 detects calcium level reduction and activates Orai2/TRPC6 channels complex. Calcium entry via TRPC6-neuronal SOCE is needed to activate CaMKII, which can phosphorylate AMPAR. This promotes AMPARs trafficking to the post-synaptic membrane, which increases membrane depolarization and enhances NMDAR’s contribution to LTP maintenance. In addition, in neuronal tumor cells TRPC6-induced calcium entry is essential for the biosynthesis of endocannabinoids (2-AG, AEA) that have anti-inflammatory properties. Supposedly, neuroprotective effects are achieved by suppressing NF-κB via activation of the PPARγ. The binding of BDNF to TrkB results in the PLC-dependent activation of TRPC6. Calcium influx through TRPC6 may also activate CREB through three signal cascades: Ras/MEK/ERK, RAS/PI3K/Akt, and CaM/CAMKIV. Neuroprotective effect is achieved via CREB-dependent activation of the BDNF transcription. Moreover, in DGC and GABAergic interneurons TRPC6 may impact on Kv4.3 translocation to the neuronal membrane via ERK-dependent signaling (indicated by a wavy line). Kv4.3 impedes fast-spiking in interneurons, and afterwards increases neuronal excitability in principal neurons. Thus, TRPC6 possibly plays a role in regulation of neuronal excitability. Finally, TRPC6-ERK1/2 neuroprotective signaling pathway has been also reported to protect DGC via facilitation of mitochondrial fission (indicated by a wavy line).

**Table 1 ijms-24-15415-t001:** Therapeutic effects of positive TRPC6 modulators demonstrated by behavioral tests.

TRPC6 Positive Modulator	Behavioral Test	Daily Dose, Method	Duration	Animal Model (Disease Model)	Therapeutic Effect	Brain Region/System Involved	Reference
Hyperforin	Morris water maze test	6 µM, intracerebral injection	4–18 days	Sprague–Dawley rats injected with Aβ (AD)	Improvement of spatial memory	Hippocampus, striatum, basal forebrain, cerebellum and cerebral cortex [105]	[92]
	Open field test	3 mg/kg, intragastric administration	14 days	Early separated from parents Wistar rats (depression)	Anxiolytic effect	Mesolimbic/nigrostriatal dopamine systems [106]	[93]
	Novelty suppressed feeding test				Anxiolytic effect and antidepressant effects	Amygdala, hippocampus [107]	
	Forced swimming test				Antidepressant effect	Amygdala, hippocampus [106]	
	Conditioned avoidance test	1.25 mg/kg, oral administration	7 consecutive days and day 17 (i.e., after 9 days without treatment)	Wistar rats (depression)	Antidepressant effect and improvement of memory	Hippocampus/amygdala [108]	[94]
	Passive avoidance tests		3 times for 1 day (1 h before, 1 h and 23 h after training)		Antidepressant effect and improvement of memory	Hippocampus/amygdala [108]	
Hyp13	Open field test	5 mg/kg, intraperitoneal injection	Once (20 min before the test)	TRPC6 KO mice (depression)	Anxiolytic effect	Mesolimbic/nigrostriatal dopamine systems [106]	[97]
	Novelty suppressed feeding test				Anxiolytic effect and antidepressant effects	Amygdala, hippocampus [107]	
	Forced swimming test				Antidepressant effect	Amygdala, hippocampus [106]	
IDN5706	Morris water maze test	2 mg/kg, intraperitoneal injection	4 weeks	APPPSEN1deltaE9 (AD)	Improvement of spatial memory	Hippocampus, striatum, basal forebrain, cerebellum and cerebral cortex [105]	[98]
AZP2006	Y-maze test	3 mg/kg, oral administration	4–8 months	C57B/6Rj mice injected with Aβ (AD)	Improvement of spatial memory	Hippocampus [106]	[99]
	Passive avoidance test				Antidepressant effect and improvement of memory	Hippocampus/amygdala [108]	
Resveratrol	Novel object recognition test	50 or 100 mg/kg, intraperitoneal injection	4 weeks	Sprague–Dawley rats fed a high-fat diet (type 2 diabetes mellitus)	Anxiolytic effect and improvement of memory	Hippocampus and perirhinal cortex [109]	[100]
	Elevated plus maze test,				Improvement of acquisition and retention memory	Amygdala [106]	
	Light–dark passive avoidance test				Improvement of emotional memory	Hippocampus/amygdala [108]	
	Radial arm maze				Improvement of working and reference memories	Hippocampus, frontal cortex, and forebrain cholinergic pathways [110]	
	Nest building				Restore of cognitive function	General (including hippocampus) [106]	
C20	Cued fear conditioning test	10 mg/kg, intraperitoneal injection	14 days	5xFAD mice (AD)	Improvement of cued memory	Amygdala [102]	[20]
	Context fear conditioning test				Improvement of context memory	Hippocampus, amygdala [111]	

## Data Availability

Not applicable.

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
