# Peer review of "Role of Neuronal TRPC6 Channels in Synapse Development, Memory Formation and Animal Behavior"

_ijms, 2023, doi:10.3390/ijms242015415_

Round 1
Reviewer 1 Report
1. The Introduction is not concise enough, it looks like a stack of many statements. No references are provided to support statements, like line 55-66.
2. For the signal pathway, the discussion is not clear and confusing. For example, the author claimed that TRPC6 play neuroprotection through blocking extrasynaptic NMDAR but did not discuss it or provide supporting references. It did not discuss the agonism of synaptic NMDAR by TRPC6 in neuronprotection as it stated in the beginning of the paragraph. Similar problem exist in endocannabinoid.
3. The manuscript did not discuss the role of TRPC6 in synapse development although it appears in the title.
4. The Figure 1 is not clear. It seems that TPRC6 is a downstream target of NMDAR rather than a modulator.
5. The manuscript requires carefully editing and improvement. For example, Line 86-97 is too long, most of the content is unrelated with TRPC6; Line 110-112 should be moved to the statement of its mechanism.
6. Abbreviation should be provided with full name at first appearance and used through all following statements, like AD (Line 62), nSOCE, AEA, OAG, etc
The re-organization of sentences and paragraph is required.
Reviewer 2 Report
Reviewer Comments
The present review article discussed the role of TRPC6 in increased synapse formation and enhanced learning and memory in animal models. TRPC6 has been shown to be involved in different synaptic-protective signaling pathways, including antagonism of NMDAR, activation of BDNF, and postsynaptic store-operated calcium entry. Overall article discussed the role of TRPC6 in memory formation in various parts of the brain to understand how TRPC6-specific pharmaceutical agents will affect memory in distinct parts of the CNS.
The research paper is well-planned however lacks current research and expert opinion. However, a few suggestions are here to incorporate.
Scientific comments
1. In Table 1, the experimental periods and doses should be included, which are very important.
2. In the abstract, we have observed ……independent fear memory is mentioned however in the main body of the article only 1 case of 5XFAD mice was discussed.
3. The major importance of this article is the role of neuronal TRPC6 channels in synapse development, memory formation, and animal behavior however research-based data were vaguely described or completely skipped.
4. Detailed mechanism of action of TRPC6 in animal learning and memory of animals has been discussed.
5. Without a more detailed explanation of current researchers etc. and an expert opinion of the importance of the results this review is mostly just information.
6. More studies with references could be added to improve the article Ex- https://doi.org/10.3389/fcell.2020.618663, https://doi.org/10.3389/fphys.2020.00238.
Minor corrections/suggestions
7. Check all references for correction (Ex. Ref no 85. Activity. Sci. Reports 2014 41 2014).
8. Line 213, in 3rd section instead of 3rd chapter.
9. Line 151, (AMPAR)), remove 1 extra bracket.
Round 2
Reviewer 1 Report
1. The revision of Introduction is still not concise enough, for example, Line 49-61 is not relevant.
2. One section discussing Synapse development should be included since it is appears in the Title.
3. Many statements are not reasonable, for example, Line 295-297, why TRPC6 modulates neuronal excitability? The Section of 4 should discuss the signaling pathway of TRPC6 in synapse development, memory formation, and behaviors.
4. In Section 3, NMDAR is discussed while it is not intracellular signaling pathway. While ERK is intracellular signaling but it is discussed in Section 4. And Kv channels mentioned in Section 4 are not included in Figure 1
5. Table 1, the subtitle of Disease model should use Animal model not Mice model
6. Abbreviation: Line 36, 47, 49 for AD. Line 263, 269, 286, 297 for DG
7. Some denotations of the different lines/arrow in Figure 1 are helpful to understanding different pathways.
Reviewer 2 Report
All the suggestions has been incorporate dby authors in revised MS. The article can be accepeted in present form.
Author Response
We thank Reviewer 2 for valluable comments and positive evaluation of our manuscript in Revision 1 process.
The Reviewer 2 did not provide further comments in Revision 2 procedure.
Round 3
Reviewer 1 Report
no further question.